# Vector-Borne Pathogens in Stray Cats in Eastern Germany (Thuringia)

**DOI:** 10.3390/ani13162574

**Published:** 2023-08-10

**Authors:** Ingo Schäfer, Axel Peukert, Katharina Kerner, Elisabeth Müller

**Affiliations:** 1LABOKLIN GmbH and Co. KG, Steubenstraße 4, 97688 Bad Kissingen, Germanymueller@laboklin.com (E.M.); 2Small Animal Practice Oberweimar, Taubacher Straße 13, 99425 Weimar, Germany

**Keywords:** feline vector-borne infection, anaplasmosis, hepatozoonosis, zoonosis

## Abstract

**Simple Summary:**

There is currently no available data on the prevalence of vector-borne pathogens (VBPs) in stray cats in Germany. In this study, clinically healthy stray cats were investigated for selected VBPs. Of the 50 stray cats included, 22% tested positive for at least one VBP by direct and 64% by indirect detection methods. Stray cats may therefore represent important pathogen reservoirs and contribute to the spread of VBPs, many of which have zoonotic potential, for example, *Bartonella* spp., *Rickettsia* spp., and *Anaplasma phagocytophilum*.

**Abstract:**

Bacterial, protozoal, and viral vector-borne pathogens (VBPs) can cause infections in cats. There is little information on feline VBP prevalence in Germany. Stray cats are frequently exposed to vectors but receive no veterinary care. The aim of this study was to determine the prevalence of selected VBPs in stray cats. EDTA blood and serum samples were taken from apparently healthy stray cats during a spay/neuter campaign in the federal state of Thuringia. Overall, 11/50 (22%) and 32/50 (64%) cats tested positive for at least one VBP by direct and indirect detection methods, respectively. PCR testing of EDTA blood detected hemotropic *Mycoplasma* spp. in 12% of cats, *Hepatozoon* spp. in 10%, and *Anaplasma phagocytophilum* in 4%. PCR testing for *Rickettsia* spp. and piroplasms was negative. IFAT on serum samples showed 46% of cats had detectable antibodies for *Bartonella* spp., 30% for *Rickettsia* spp., and 16% for *A. phagocytophilum*. The cats were additionally tested for feline coronavirus, FIV, and FeLV to identify potential risk factors for pathogen contact and/or infections. No correlation between FIV and FeLV status and VBP positivity was detected. *Anaplasma phagocytophilum*, *Rickettsia* spp., and *Bartonella* spp. have zoonotic potential, and surveillance is recommended in the context of the One Health approach.

## 1. Introduction

Vector-borne pathogens (VBPs) have become increasingly important to veterinary medicine due to changes in climate and a surge in pet travel and import. Additionally, domestic and international travel and trade carry the risk of vector introduction into currently non-endemic areas in Europe. Several bacterial, protozoan, helminthic, and viral VBPs are known to infect cats in Central Europe, all of which are mainly transmitted by blood-sucking arthropods. *Anaplasma* (*A*.) *phagocytophilum* is primarily transmitted by *Ixodes* (*I*.) *ricinus* ticks [1]. Fleas and other blood-sucking arthropods may be potential vectors for hemotropic *Mycoplasma* spp. [2,3]. *Rickettsia* (*R*.) *felis* is the most important *Rickettsia* sp. in cats in Germany and is transmitted by *Ctenocephalides felis* and *Archaeopsylla erinacei* [4]. The most important vector for *Bartonella* spp. is *Ctenocephalides felis* [5], though *Bartonella henselae* has been detected in *I. ricinus* ticks [6]. *Cytauxzoon* sp. is transmitted by ticks, likely *Dermacentor* spp. and/or *I. ricinus* [7]. Fleas, lice, and ticks are suspected vectors for *Hepatozoon* (*H*.) spp., of which, *H. felis*, *H. canis*, and *H. silvestris* are causing infections in cats [8,9,10,11,12,13,14]. Additionally, transplacental infections of *H. felis* may also be possible in cats [8] as such cases regarding *H. canis* have been confirmed in dogs [15]. It is suspected that *Candidatus* Mycoplasma haemominutum may be transmitted horizontally [16], and *Candidatus* Mycoplasma turicensis could be transmitted via bites and cat fights [17].

Stray cats usually do not have any veterinary care or ectoparasite prophylaxis, also vector exposure may be much more frequent than in their domestic counterparts. Previous studies on feline VBPs predominantly focused on domestic cats and may therefore not be representative of the stray population, as VBP prevalence could differ significantly. Infections with the following bacterial pathogens diagnosed by PCR have been reported in domestic cats: *A. phagocytophilum* in 0.3–4% of tested cats and as individual case reports [18,19,20,21,22], hemotropic *Mycoplasma* spp. with predominantly *Candidatus* Mycoplasma haemominutum in 7.2–22.5% of tested cats [21,23,24,25], and *Bartonella* spp. in 0–16% of tested cats [21,26,27,28]. There are no reports on the prevalence of *R. felis* in domestic cats in Germany. Regarding protozoal pathogens, a single report of an infection with *Cytauxzoon* sp. in a cat without a travel history in the federal state of Saarland in southern Germany has been published, which is the first autochthonous infection described in Germany [29]. There are individual case reports of feline autochthonous infections with *Hepatozoon* spp. in Central Europe, namely of *H. felis* in a cat from Austria [30] and of *H. silvestris* in a cat from Switzerland [12]. One study detected *H. felis* in sixteen cats living in Germany, all of which had a travel history [31].

There are limited data on wildcats: one study examined blood and splenic tissue of 96 European wildcats (*Felis silvestris silvestris*) between 1998 and 2020 and reported infections with *Cytauxzoon europaeus* (45/96 cats, 46.9%), *H. silvestris* (34/96 cats, 35.4%), *Candidatus* Mycoplasma haemominutum (7/96 cats, 7.3%), *H. felis* (6/96 cats, 6.3%), *Bartonella* spp. (3/96 cats, 3.1%), and *Mycoplasma ovis* (1/96, 1%) [32].

Immunosuppression, especially caused by feline leukemia virus (FeLV) and feline immunodeficiency virus (FIV), not only predisposes cats to infections with VBPs but also increases the rate of clinical manifestations in infected cats, as has been described in leishmaniosis [33], hepatozoonosis [34,35,36], and hemotropic mycoplasma [37,38,39,40,41].

There is currently no information on VBPs in stray cats in Germany. The aim of this study was to investigate stray cats caught as part of a spay/neuter campaign in the federal state of Thuringia in eastern Germany for several VBPs in order to determine VBP prevalence in this population.

## 2. Materials and Methods

Stray cats were caught in 14 different locations in the German federal state Thuringia and transported to the small animal practice Oberweimar (Weimar, Germany) as part of a spay/neuter program authorized by the local government (F-TIER220034). Ethylenediaminetetra-acetic acid (EDTA) blood and serum samples were taken prior to anesthesia to timely detect life-threatening hematological and biochemistry abnormalities. The samples were submitted by the practice to the commercial laboratory LABOKLIN (Bad Kissingen, Germany) between April and September 2022. Hematological analysis was performed on the Sysmex XN-V (Sysmex Deutschland GmbH, Norderstedt, Germany) and a routine biochemistry analysis on the Cobas 8000 (Roche Diagnostics Deutschland GmbH, Mannheim, Germany). 

Wildcats were excluded from this study, as they were identified by their unique signalment and were not captured during the initial spay/neuter campaign. This study included only previously intact cats with unremarkable clinical examination. 

Direct detection methods for selected VBPs included PCR or antigen testing on surplus sample material following hematology and biochemistry testing. TaqMan^®^ real-time PCR testing was performed on deoxyribonucleic acid isolated from EDTA blood samples for *A. phagocytophilum*, *Rickettsia* spp., hemotropic *Mycoplasma* spp., piroplasms, and *Hepatozoon* spp. (Table 1). PCR results were evaluated on a qualitative basis (negative/positive only) with a cut-off Ct value of 35. Each PCR run included a negative and a positive control as well as an extraction control for each sample to confirm nucleic acid extraction and exclude PCR inhibition (DNA Process Control Detection Kit, Roche Diagnostics Deutschland GmbH, Mannheim, Germany). EDTA blood was also used for antigen testing for FeLV (NovaTec VetLine Feline Leukemia Virus Antigen ELISA, NovaTec Immundiagnostica, Dietzenbach, Germany).

Sera were used for antibody testing. As an indirect detection method, immunofluorescence antibody testing (IFAT) was performed for *A. phagocytophilum* (MegaFLUO ANAPLASMA; MegaCor Diagnostik GmbH, Lindau, Germany), *Rickettsia* spp. (RICKETTSIA CONORII IFA SLIDE, Viracell, Granada, Spain), and *Bartonella* spp. (MegaFLUO BARTONELLA henselae; MegaCor Diagnostik GmbH, Lindau, Germany) in accordance with manufacturer guidelines. Antibody ELISA was used to test for FIV (NovaTec VetLine Feline Immunodeficiency Virus ELISA, NovaTec Immundiagnostica, Dietzenbach, Germany) and feline coronavirus (NovaTec VetLine Feline Corona Virus (FCoV/FIP) ELISA, NovaTec Immundiagnostica, Dietzenbach, Germany).

According to the terms and conditions of the laboratory LABOKLIN as well as the decision of the government of Lower Franconia RUF-55.2.2.2532-1-86-5, neither permission from animal owners nor approval by the animal welfare commission is necessary for additional testing on residual sample material once diagnostics are completed.

Descriptive statistical analysis was performed using SPSS for Windows (version 28.0; IBM). *p* < 0.05 was considered statistically significant. Shapiro–Wilk test was used for assessment of normal distribution. Kruskal–Wallis test was used to calculate statistical significance between the study groups. Bonferroni correction was applied where necessary. Binary logistic regression analyses (bivariate) were performed to determine the effect of sex, age, and FIV antibody presence on PCR results.

## 3. Results

Of the 50 stray cats included in this study, 18 (36.0%) were male and 32 (64.0%) were female. An approximate age was documented in 48/50 cats, estimated by the veterinarian based on general and dental examination findings (96.0%, median 3.0 years, standard deviation 2.1 years, minimum 0.3 years, maximum 8 years). Rectal temperature (median 38.5 °C, standard deviation 0.5 °C, minimum 37.3 °C, maximum 39.4 °C) and bodyweight (median 3.0 kg, standard deviation 1.0 kg, minimum 1.3 kg, maximum 5.6 kg) were documented for all cats. The stray cats were caught in 15 different locations surrounding the Weimarer Land region in Germany (Ettersburg *n* = 10, Niederreißen *n* = 10, Großobringen *n* = 5, Hohenfelden *n* = 4, Tonndorf *n* = 4, Lehnstedt *n* = 3, Niedergrundstedt *n* = 3, Weimar-Schöndorf *n* = 3, Meckfeld *n* = 2, Bucha *n* = 1, Daasdorf *n* = 1, Guthmannshausen *n* = 1, Heichelheim *n* = 1, Holzdorf *n* = 1, Lengefeld *n* = 1).

The results of direct and indirect detection methods for the selected VBPs can be seen in Table 2. PCR testing was positive for at least one VBP in 11/50 cats (22.0%), most notably for hemotropic *Mycoplasma* spp. (6/50 cats, 12.0%, all *Candidatus* Mycoplasma haemominutum), *Hepatozoon* spp. (5/50 cats, 10.0%), and *A. phagocytophilum* (2/50 cats, 4.0%). More than half of the cats in this study had antibody titers for at least one pathogen (33/50 cats, 66%) (Table 2). 

The multiple logistic regression model was statistically significant in cats with *A. phagocytophilum* antibodies detected by IFAT (constant: B = −4.397, SE = 1.491, Wald = 8.692, *p* = 0.003). Cats older than 3 years had 10.908 times greater odds of antibody titers than their younger counterparts (95% confidence interval 1.607; 74.506) (Table 3).

Biochemistry results were available for all cats in this study, while hematology results were available for all save one cat (2%). The most common hematological abnormality was leukocytosis (24/49 cats, 49.0%), most often in combination with eosinophilia (24/49 cats, 49.0%) and neutrophilia (14/49 cats, 28.6%). The most frequent biochemical abnormalities were elevated creatine kinase (36/50 cats, 72.0%), hyperglycemia (28/50 cats, 56.0%), and hyperphosphatemia (25/50 cats, 50.0%) (Appendix A). 

PCR results did not have any statistically significant impact on hematocrit, white blood cell count, or platelet count (all *p* > 0.05). Regarding biochemistry, there was a statistically significant difference in alanine transaminase (ALT) levels in cats with *Hepatozoon* spp. infection and those with negative PCR (*p* = 0.036, Appendix A).

There was one case of mild anemia (hematocrit 29%) in a cat with a coinfection of *A. phagocytophilum* and *Hepatozoon* spp. Furthermore, mild leukocytosis was detected in three cats with *Candidatus* Mycoplasma haemominutum infections (11.1–16.8 × 10^9^/L) and one cat infected with *Hepatozoon* spp. (17.3 × 10^9^/L). There were no cases of thrombocytopenia in cats with positive PCR of any VBP (Appendix A).

*A. phagocytophilum* was detected by PCR in two cats, both of which also showed positive antibody titers for the pathogen (1:40, 1:320). The first cat was caught in Ettersberg. In addition to *A. phagocytophilum*, this cat also tested positive for *Hepatozoon* spp. by PCR and had antibodies to *R. felis* (1:128) and mild non-regenerative anemia (hematocrit 29%). The second cat was caught in Nierderreißen. It additionally tested positive for *Candidatus* Mycoplasma haemominutum and showed mild lymphopenia. Both cats were hyperglycemic and had mildly elevated creatine kinase levels.

Five cats had positive PCR results for *Hepatozoon* spp, all of which had been caught in Ettersburg. One additionally tested positive for *A. phagocytophilum*, as mentioned above. Three cats (60.0%) had FIV antibody titers. Serology showed antibodies to *R. felis* in 3/5 cats (60.0%; 1:128 *n* = 2, 1:256 *n* = 1) and to *A. phagocytophilum* also in 3/5 cats (1:40 *n* = 1, 1:320 *n* = 2). One cat (20.0%) was mildly anemic, and another (20.0%) had mild leukocytosis with neutrophilia. Two cats (40%) had mildly elevated creatine kinase levels (201 and 171 mmol/L).

Six cats had positive PCR results for *Candidatus* Mycoplasma haemominutum. Among them, serology revealed FIV antibodies in two cats (33.3%), *Bartonella* spp. antibodies in two cats (33.3%, 1:40, and 1:320), *A. phagocytophilum* antibodies in two cats (33.3%, 1:40 and 1:320), and *R. felis* antibodies in one cat (16.7%, 1:128). Three cats (50.0%) had mild leukocytosis, one of which also had mild lymphocytosis and eosinophilia. One cat (16.7%) had mild thrombocytosis. None of these cats were anemic. All six cats were caught in different locations.

FIV antibodies were detected in 12/50 cats (24.0%; Table 1). Among them, five (41.7%) tested positive for at least one VBP by PCR (*Hepatozoon* spp. *n* = 3; *Candidatus* Mycoplasma haemominutum *n* = 2). Serology showed 9/12 cats (75.0%) had antibodies to feline coronavirus, six (50.0%) to *Bartonella* spp., four (33.3%) to *A. phagocytophilum*, and three (25.0%) for *Rickettsia felis*. None of the twelve cats were anemic, but six (50.0%) had eosinophilia, five (41.7%) had mild leukocytosis, and one cat (8.3%) was mildly thrombocytopenic. 

Two cats presented with mild anemia, one of which has been described above. The second cat had a hematocrit of 24%, as well as moderate leukocytosis with neutrophilia, lymphocytosis, monocytosis, and no signs of regeneration. No VBPs were detected in this cat.

## 4. Discussion

This is the first study to evaluate clinically healthy stray cats affected by selected VBPs in Germany. Sharing environment may prove to be an important link between the domestic cats and local wildlife. Such comparisons between these two groups are therefore of great interest not only to veterinary clinical practice but also to public health in both short and long term. The importance of wildlife, dogs, and horses as pathogen reservoirs has been well documented, but the significance of cats in the epidemiology of several VBPs is still largely unknown [42].

Wildcats have similar living conditions compared to stray cats and under normal circumstances do not receive any veterinary care or ectoparasite prophylaxis. A study with 96 wildcats in Germany reported VBPs in 67/96 cats (70%) tested by PCR on spleen aspirates and whole blood [32] with a spectrum of pathogens analyzed that was comparable to our study. The pathogens detected included *Cytauxzoon europaeus* (*n* = 45), *Hepatozoon* spp. (*n* = 40), *Candidatus* Mycoplasma haemominutum (*n* = 7), and *Bartonella* spp. (*n* = 3), while PCR testing for Anaplasmataceae and Rickettsiales was negative in all wildcats [32]. The higher overall rate of positive PCR results compared to our study (22%) might be explained by the fact that the study on wildcats did not include a health screening and therefore would have included cats with clinical manifestations, which were excluded in our study accordingly, as only cats classified as clinically healthy underwent surgery. It should be noted that the infection rate with *Hepatozoon* spp. (42%, predominantly *H. silvestris* (*n* = 34) and *H. felis* (*n* = 6)) was much higher in this study from 2022 [32] than in our study of stray cats (10%, no sequencing performed). Species differentiation was not available due to the small volumes obtained and might have revealed an explanation for the different rates of positivity. All affected stray cats in this study were caught in the same location, which may indicate a pronounced regionality of pathogen distribution. While piroplasms were not detected in any cats of this study, two previous studies in wildcats from Germany reported *Cytauxzoon europaeus* infection rates of 4% [32] and 65% [43], respectively. In this case, the difference in infection rates may be due to absence of competent vectors and/or the pathogen itself in the geographical area of our study or due to potential differences in susceptibility for this VBP in wildcats, domestic cats, and stray cats.

Foxes have been established as a crucial contributor to VBP transmission in Central Europe in recent years. In a study of red foxes (*Vulpes vulpes*) from the Czech Republic, 94% tested positive by PCR for at least one of seven tick-borne pathogens, including *H. canis* (81%) and *A. phagocytophilum* (3%) [44]. In a German study examining splenic tissue from red foxes for *Hepatozoon* spp., 45% tested positive [45]. In Italy, 5.2% of red foxes tested had detectable antibodies to *A. phagocytophilum* [46]. These studies did not discriminate between healthy animals and those with clinical manifestations, which may explain the higher rate of positive PCR results. Serology results, however, were more comparable. 

In stray dogs from Southern Italy, *Anaplasma* spp. seroprevalence was reported to be 7.8% [47], a rate much lower than in the cats of this study (16%). It should be noted that *I. ricinus* ticks, a crucial vector for *A. phagocytophilum*, are more prevalent in Northern Italy and that the study in question used different diagnostic assays. Both factors directly impair the comparison of data and may explain the discrepancy of only 5% of dogs in Germany testing positive by PCR while 27% tested positive by serological testing [48] the same was observed for domestic cats in Germany where 4% tested positive by PCR and 23% by IFAT respectively [18]. These numbers are consistent to the findings of this study where 4% of stray cats tested positive by PCR and 23% showed increased antibody titers for *A. phagocytophilum*. 

One previous study in domestic cats in Germany reported infection rates of 9% for *Hepatozoon* spp. (PCR) and 11% for *Rickettsia* spp. (IFAT) [49], both linked to travel history. In a further study of domestic cats from Germany, *Heptozoon* spp was detected in 7% by PCR [31]. While it is impossible to ascertain any travel history in strays, both studies had comparable prevalence rates to our study regarding stray cats (10%). The seroprevalence of *Bartonella* spp. in Europe varies greatly in the literature (8–81%), likely depending on chosen area and corresponding to flea prevalence [50]. Regarding domestic cats in Germany, the antibody prevalence is overall lower, ranging from 0% to 16% respectively [21,26,27,28]. The stray cat seroprevalence our study is remarkably higher, reaching 46%, most likely due to higher exposure to the main vector *Ctenocephalides felis* [5]. Even higher seropositivity was seen in a colony of stray cats from Spain (81%) and cats having an owner [51]. Cats are often subclinical carriers of *Bartonella* spp. and represent their principal reservoir [50]. Feral cats may therefore represent an important zoonotic pathogen reservoir, as *Bartonella henselae* may cause cat scratch disease in humans. There is limited knowledge about the prevalence of *A. phagocytophilum* in domestic cats in Germany, though it has been reported that the vector most frequently found on domestic cats is the *I. ricinus* tick (91%) [52], a well-established vector for *A. phagocytophilum*. A total of one study demonstrated an infection rate of 3% (18/619 cats) and antibody presence in 23% (191/847) of domestic cats [18], which corresponds to our findings (positive PCR in 4%, positive IFAT in 16%). Both cats with positive PCR results also had detectable antibody titers, which is consistent with the literature [18,53]. Fever is one of the most common clinical signs in *A. phagocytophilum* infections [53], while only clinically healthy cats based on a general examination were included, therefore acute infections were probably underrepresented in this cohort. In domestic cats in Germany, infections with *Hepatozoon* spp. are usually associated with a travel history to endemic countries [49]. However, individual autochthonous infections have been reported in Austria (*H. felis*) [30] and Switzerland (*H. silvestris*) [12]. Transplacental infections may occur in dogs with *H. canis* [15], though this transmission route has not been documented in cats [8]. It is very likely that the stray cats in this study were infected in Germany. Hepatozoonosis in domestic cats is strongly associated with outdoor access [8], which may explain the higher prevalence in wildcats and stray cats. Regarding piroplasms, a study of 552 cats, 65% of which were from Germany, did not report any *Cytauxzoon* sp. Infections [54], while another study reported a single autochthonous infection with *Cytauxzoon* sp. In the southern German federal state of Saarland [29]. The results of our study in stray cats are therefore most consistent with what has been documented in domestic cats, as piroplasm infections in cats are overall rare occurrence in Europe. 

*Rickettsia felis* is the most important rickettsial species in cats in Germany where it is transmitted by *Ctenocephalides felis* and *Archaeopsylla rinaceid* [4]. Knowledge of rickettsial infections in cats is limited. DNA of *Rickettsia felis* has so far been detected in feline samples of skin and gingiva but never in peripheral blood [55]. As EDTA blood is considered suboptimal for direct detection of *Rickettsia felis*, the true infection rate in the stray cats examined may be underrepresented. 

As many of the relevant vectors show pronounced seasonality, the time of the year chosen for sampling may influence PCR detection rates. Nevertheless, it is remarkable that 22% of stray cats tested positive by PCR and 66% had positive serology for at least one VBP. Positive PCR results are largely indicative of an acute infection, while antibody detection could also show past contact with the pathogen. 

VBP occurrence is generally associated with the distribution of vectors, which can be influenced by changes in climate, land use, human density, and pathogen reservoirs. In general, immunosuppression, especially caused by the feline leukemia virus (FeLV) and feline immunodeficiency virus (FIV), not only predisposes cats to infections with VBPs but also increases the rate of clinical manifestations in already infected cats. This has been described for cats with leishmaniosis [33], hepatozoonosis [34,35,36], and infections of hemotropic *Mycoplasma* spp. [37,38,39,40,41]. Multiple logistic regression analysis did not show any statistically significant impact of FIV status on positive PCR results in this study. However, the selection criteria may have influenced the statistical analysis, as our cohort only included clinically healthy cats. No FeLV-positive cats were included in this study, therefore no effect of FeLV status on diagnostic test results could be investigated. 

Stray cats older than three years had more than ten times higher odds of testing positive for *A. phagocytophilum* than their younger counterparts (OR 10.908, 95% CI [1.607; 74.056]), which can be explained by the cumulative exposure risk that comes with older age. While not statistically significant, a similar trend was seen for *Rickettsia* spp. (OR 2.304, 95% CI [0.566; 9.389]) and *Bartonella* spp. (OR 1.116, 95% CI [0.299; 4.170]).

All six stray cats testing positive for hemotropic *Mycoplasma* spp. in this study were infected with *Candidatus* Mycoplasma haemominutum. None of these animals were anemic, while three cats showed mild leukocytosis. This is consistent with the literature reporting no significant hematological abnormalities in majority of cats infected with *Candidatus* Mycoplasma haemominutum [56]. Two out of six infected stray cats in this study (33%) additionally tested positive for FIV antibodies. Previous studies are divided on the role of FIV in hemotropic mycoplasma infections in cats [37,38,39,40,41].

There appears to be no association between *Hepatozoon* spp. infections and either gender or age of the cats [8], however, studies are divided on the impact of FIV and FeLV infection [34,35,36]. Three out of five stray cats (60%) with *Hepatozoon* spp. infections also tested positive for FIV, and one more cat showed an inconclusive result. This indicates that immunosuppression may be a predisposing factor to *Hepatozoon* spp. infection, though there was no statistically significant impact on diagnostic test results in this study. However, this could be explained through the low numbers of feral cats included in this study. Further research may be of interest.

The small cohort of this study limits the value of the correlation between hematology or biochemistry findings and VBPs, especially for *Hepatozoon* spp., hemotropic *Mycoplasma* spp., and coinfections. However, there was a statistically significant difference in the alanine transaminase (ALT) levels between cats with positive and negative PCR results for *Hepatozoon* spp. An underlying cause for ALT elevations in *Hepatozoon* spp. infections is not known, but this finding could also be related to pathological changes not evident during general examination.

Thrombocytopenia is the most diagnostically relevant hematological finding in cats with *A. phagocytophilum* infections, followed by anemia and leukopenia [53]. Neither of the two stray cats infected with *A. phagocytophilum* in this study was thrombocytopenic. One cat coinfected with *Hepatozoon* spp. and *A. phagocytophilum* did show mild anemia, which is consistent with the literature [53]. In fact, domestic cats infected with *Hepatozoon* spp. often demonstrate no hematological abnormalities, as shown in a previous study where 50% of cats (6/12) with positive PCR results showed unremarkable hematology results [31]. In our study two of five stray cats (40%) with positive *Hepatozoon* PCR showed mild hematological abnormalities, while the remainder was unremarkable. 

The most remarkable biochemical abnormalities noted were hyperglycemia and elevations in creatine kinase. The hyperglycemia was probably observed due to the stress of handling the feral cats in the practice. Elevations in creatine kinase often indicate muscular trauma, which in this case, was likely related to catching and handling the animals. It is therefore quite unlikely that any VBP infections were the cause of these biochemical anomalies. Still, it should be noted that elevations in creatine kinase may be a manifestation of feline heaptozoonosis [35]. The tropism of *Hepatozoon* spp. for muscle tissues was confirmed for *H. felis* [8] and *H. silvestris* infections [12] but is currently unknown for *H. canis* infections in cats. In the feral cats, muscular damage due to handling and/or due to the infection with *Hepatozoon* spp. was possible.

This study indicates that stray cats in Germany are quite commonly exposed to various VBPs. These stray cats share environment with wildcats and domestic cats with outdoor access, causing a risk of infection across all three groups. Additional studies are necessary to clarify potential routes of transmission, the role of cats in these transmission cycles, and the clinical impact of VBPs on cats in general. All cats in this study were asymptomatic, and even in the infected cats, hematological anomalies were rare. Considering the zoonotic potential of *A. phagocytophilum*, *Rickettsia* spp., and *Bartonella* spp. and their importance in the One Health approach, the possibility of subclinical infections with these pathogens in stray cats is of concern. We could not demonstrate a link between immunosuppression caused, e.g., by FeLV or FIV, and VBPs in this study. The high exposure rates to at least one of the seven VBPs reveal a high frequency of occurrence and thus is of potentially high significance within the One Health concept.

## Figures and Tables

**Table 1 animals-13-02574-t001:** TaqMan^®^ real-time PCR testing applied for the detection of vector-borne pathogens performed on stray cats in eastern Germany (Thuringia).

Pathogen	Gene Target	Primer Sequence (5′-3′)
*Anaplasma phagocytophilum*	60 kDa heat shock protein	F: CTCTGAGCACGCTTGTACT R: GCCTTTACAGCAGCAACTTGAAG
*Rickettsia* spp.	23S rRNA	F: AGCTTGCTTTTGGATCATTTGG R: TTCCTTGCCTTTTCATACATCTAGT
*Mycoplasma haemofelis*	16S rDNA	F: GTGCTACAATGGCGAACACA R: TCCTATCCGAACTGAGACGAA
*Candidatus* Mycoplasma hemominutum	16S rDNA	F: TGATCTATTGTKAAAGGCACTTGCT R: TTAGCCTCYGGTGTTCCTCAA
*Candidatus* Mycoplasma turicensis	16S rDNA	F: AGAGGCGAAGGCGAAAACT R: CTACAACGCCGAAACACAAA
Piroplasms (*Babesia* spp./*Cytauxzoon* sp.)	small subunit ribosomal DNA	F: AATACCCAATCCTGACACAGGG R: TTAAATACGAATGCCCCCAAC
*Hepatozoon* spp.	18S rRNA	F: AACACGGGAAAACTCACCAG R: CCTCAAACTTCCTCGCGTTA

**Table 2 animals-13-02574-t002:** Percentages of stray cats in Thuringia (Germany) tested for several vector-borne pathogens by direct and/or indirect detection methods (*n* tested positive/N total (%)).

Pathogen	Direct Detection Methods (PCR)	Indirect Detection Methods (IFAT)
*Anaplasma phagocytophilum*	2/50 (4)	8/50 (16)
*Rickettsia* spp.	0/50 (0)	15/50 (30)
Hemotropic *Mycoplasma* spp.	6/50 (12)	-/-
*Bartonella* spp.	-/-	23/50 (46)
Piroplasms (*Babesia* spp./*Cytauxzoon* sp.)	0/50 (0)	-/-
*Hepatozoon* spp.	5/50 (10)	-/-
Total (including coinfections)	11/50 (22)	33/50 (66)

IFAT = immunofluorescence antibody testing, PCR = polymerase chain reaction.

**Table 3 animals-13-02574-t003:** Multiple logistic regression analysis in 48 stray cats tested positive for several vector-borne pathogens by direct (polymerase chain reaction [PCR]) and/or indirect (immunofluorescence antibody test [IFAT]) detection methods in which data regarding sex, age, and FIV antibody status were available.

Pathogen	Variables	*B*	*SE*	Wald	*p*	Odds Ratio	95% CI for Odds Ratio (Lower/Upper Bound)
*Anaplasma phagocytophilum* PCR	Sex (male)	−0.930	1.510	0.379	0.538	0.395	0.020/7.618
	Age (>3 years)	1.482	1.512	0.961	0.327	4.402	0.227/85.233
	FIV (positive)	18.806	10990.485	0.000	0.999	-	-
*Anaplasma phagocytophilum* IFAT	Sex (male)	0.1685	1.185	2.022	0.155	5.493	0.528/55.028
	Age (>3 years)	2.390	0.977	5.978	0.014	10.908	1.607/74.065
	FIV (positive)	1.894	1.022	3.436	0.064	6.643	0.897/49.162
*Rickettsia* spp. IFAT	Sex (male)	0.573	0.708	0.655	0.418	1.774	0.443/7.106
	Age (>3 years)	0.835	0.717	1.357	0.244	2.304	0.566/9.389
	FIV (positive)	−0.275	0.787	0.122	0.727	0.759	0.162/3.552
*Bartonella* spp. IFAT	Sex (male)	0.247	0.607	0.166	0.684	1.281	0.390/4.206
	Age (>3 years)	0.110	0.673	0.027	0.870	1.116	0.299/4.170
	FIV (positive)	0.141	0.680	0.043	0.836	1.151	0.304/4.206
*Candidatus* Mycoplasma haemominutum PCR	Sex (male)	−1.345	0.939	2.051	0.152	0.260	0.041/1.642
	Age (>3 years)	0.378	0.991	0.145	0.703	1.459	0.209/10.171
	FIV (positive)	0.183	1.002	0.033	0.855	1.201	0.168/8.558
*Hepatozoon* spp. PCR	Sex (male)	−0.767	1.021	0.564	0.453	0.465	0.063/3.437
	Age (>3 years)	0.513	1.058	0.235	0.628	1.670	0.210/13.286
	FIV (positive)	−1.558	1.017	2.347	0.125	0.211	0.029/1.545

*B* = unstandardized regression weight; CI = confidence interval, *SE* = standard error of the mean, Wald = test statistic for Wald chi-squared test. Variables entered: regarding age: >3 years, regarding sex: male, regarding FIV status: positive. Degrees of freedom were 1 for all Wald statistics.

## Data Availability

Not applicable.

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
