# Peer review of "Vector-Borne Pathogens in Stray Cats in Eastern Germany (Thuringia)"

_animals, 2023, doi:10.3390/ani13162574_

Round 1

Reviewer 1 Report

In this work, the authors describe the prevalence of different vector-borne diseases in stray cats in Germany. The authors also evaluate some predisposing factors (other viral infections) and the hematological and biochemical alterations of the infected animals. Although the amnoscript contains some important information that enriches the data in the literature, it cannot be considered for publication. The manuscript, in fact, requires major revisions. The English is lacking, and some sentences are not well connected to each other, making the sections confusing and fragmentary.he methodology requires some elucidation on the part of the authors. The results would require being organized in a different way, as would the discussions, which are lacking in comparison with other studies and are too long and dispersive. For the following manuscript, I recommend major revision and editing of the English, possibly to be done after a further round of revisions.

Title: The information about the use of co-infection (FIV, FeLV) as a risk factor or the hematological and biochemical findings is not of sufficient importance to be included in the title. Please have the authors change it to "Selected vector-borne diseases in feral cats in eastern Germany (Thuringia)" or something similar.

Lines 14–15: This sentence is wrong; please delete it. Searching PubMed for "cat germany vector you to numerous scientific papers, some of which are cited by the same authors.

Line 19: The term "vector-borne infectious pathogens" does not exist; please delete it throughout the manuscript.

Lines 21–24: repetition

Line 25: Reporting in the abstract how the animals were sampled is out of place.

Lines 28–31: Confusing results Please rephrase it.

Lines 34–38: These conclusions (written in unintelligible English) are not the result of this work and are misleading. The authors did not evaluate whether cats are reservoirs of these specific infections and did not clarify the epidemiological role of cats in these infections. They have only identified the DNA of these pathogens in cats after assessing risk factors. The same goes for sentences related to climate change. Authors should dwell on their own results, commenting on them and drawing conclusions about them.

Lines 45–47: References to "rabies" and "wildlife" poorly understood. Please, delete it.

Lines 50–52: Use pathogen names appropriately.

Line 62: The authors state "In general, feline vector-borne pathogens were so far predominantly investigated in domestic cats in Germany". This is in contrast with what was stated in the abstract.

Lines 64-77: The results of previous studies cannot simply be listed. Sentences of complete meaning must be enunciated in order to include such information and provide the reader with sufficient background to understand where the work that the authors have done is introduced.

Lines 79-81: As 64–77

Line 83: "To the best of the authors’ knowledge", please delete it throughout the manuscript.

Lines 95-96: Not significant for this study.

Lines 104-end of section: The authors cannot limit themselves to listing the tests used by inserting citations. They must accurately explain what they did so that other people can do the same things again.

Lines 129-132: Where are these results counted? Why use this test and not a chi square?

Lines 135-144: Some information is irrelevant.

Table 1: Results regarding the prevalence of FIV, FelV and FIP should not be included in this table, as they represent a risk/predisposing factor and the evidence of exposure to these infections is not the reason for the study.

Table 2: The table entries "increased" and "decreased" confuse the reader and are not helpful for a better understanding of the table.

Lines 317-323: MDPI journals do not have a "Limitations of the study" entry. Please explain these limits in the discussion section.

The discussion is very problematic. The results obtained without a critical view of what was observed and without comparing the results obtained with those of other studies performed in cats, wild felids, or dogs using similar or different approaches. Also, the discussion is very long, and many sentences can be completely eliminated as repetitive. The authors should also discuss the epidemiological role of cats and carnivores in general, which they may have for human infection (it is still not clear to the scientific community whether they are sentinels or reservoirs). Below, some useful works, in my opinion, to improve the discussion.

doi: 10.3390/ani11010009

doi: 10.1016/j.ttbdis.2022.102076

doi: 10.3389/fvets.2021.803424

doi: 10.1016/j.vprsr.2023.100857

Did the authors try to sequence any samples? If not, why hasn't it been done? Is other animal data available? Why not carry out a risk factor analysis? What type of sampling or sampling formula was ultimately applied?

I recommend editing of the English, possibly to be done after a further round of revisions.

Author Response

Answer to the editorial and reviewers‘ comments

19-July-2023

Dear Gray, dear reviewers,

thank you for giving us the opportunity to submit a revised draft of our manuscript titled “Vector-borne pathogens in stray cats in eastern Germany (Thuringia)” (ID: animals-2510704) to the Journal “Animals”. We appreciate the time and effort that you and the reviewers have dedicated to providing valuable feedback on our manuscript. We are grateful to the reviewers for their insightful comments, which have improved the paper. We have incorporated changes that reflect all the suggestions provided by the reviewers. All changes are highlighted in red in the revised manuscript. Please see below our point-by-point responses to the reviewers` comments. If any responses may remain unclear or if you wish additional changes, please do not hesitate to let us know.

General comments

According to the suggestions of the academic editor, we changed the term “Feral cat” to “Stray cat” throughout the whole manuscript.

The manuscript was reviewed language wise by a native English speaker, as suggested by reviewer #1 and #2.

We fully agree to both reviewers, that the manuscript needed intense revision and restructuring of the “Result”- and the “Discussion”-sections.

Reviewer #1

In this work, the authors describe the prevalence of different vector-borne diseases in stray cats in Germany. The authors also evaluate some predisposing factors (other viral infections) and the hematological and biochemical alterations of the infected animals. Although the manuscript contains some important information that enriches the data in the literature, it cannot be considered for publication. The manuscript, in fact, requires major revisions. The English is lacking, and some sentences are not well connected to each other, making the sections confusing and fragmentary. The methodology requires some elucidation on the part of the authors. The results would require being organized in a different way, as would the discussions, which are lacking in comparison with other studies and are too long and dispersive. For the following manuscript, I recommend major revision and editing of the English, possibly to be done after a further round of revisions.

Title: The information about the use of co-infection (FIV, FeLV) as a risk factor or the hematological and biochemical findings is not of sufficient importance to be included in the title. Please have the authors change it to "Selected vector-borne diseases in feral cats in eastern Germany (Thuringia)" or something similar.

We agree to this comment and changed the title according to the reviewer’s suggestion.

Title: “Vector-borne pathogens in feral cats in eastern Germany (Thuringia)”

 Lines 14–15: This sentence is wrong; please delete it. Searching PubMed for "cat germany vector you to numerous scientific papers, some of which are cited by the same authors.

We do not agree to this comment. To the best of our knowledge there are not studies investigating the prevalence of vector-borne pathogens in stray cats in Germany. Stray cats cannot be mixed up with domestic cats, as their lifestyle and potential influence as pathogen reservoirs is completely different. For example, stray cats most likely lack veterinary care and ectoparasite contact, and may have closer contact with other vectors and/or pathogen reservoirs, as e. g. ticks, fleas, and/or rodents. We clarified this in the discussion-section of the manuscript.

Line 19: The term "vector-borne infectious pathogens" does not exist; please delete it throughout the manuscript.

Has been changed throughout the manuscript.

Lines 21–24: repetition

Has been revised. In general, the abstract was intensively reviewed and several changes were performed to improve the quality (Line 18 ff).

Line 25: Reporting in the abstract how the animals were sampled is out of place.

Can you please specify this comment? In our point of view, the description of the study population represents a major part of the abstract.

Lines 28–31: Confusing results Please rephrase it.

Has been revised.

Line 23 ff: “PCR testing showed 12% of the cats were positive for hemotropic Mycoplasma spp., 10% for Hepatozoon spp., and 4% for Anaplasma phagocytophilum. PCR testing for Rickettsia spp. and piroplasms was negative.”

Lines 34–38: These conclusions (written in unintelligible English) are not the result of this work and are misleading. The authors did not evaluate whether cats are reservoirs of these specific infections and did not clarify the epidemiological role of cats in these infections. They have only identified the DNA of these pathogens in cats after assessing risk factors. The same goes for sentences related to climate change. Authors should dwell on their own results, commenting on them and drawing conclusions about them.

Has been revised.

Line 22 ff: “Cats with unremarkable general examination were included. In total, 11/50 (22%) and 32/50 (64%) cats tested positive for at least one VBP. PCR testing showed 12% of the cats were positive for hemotropic Mycoplasma spp., 10% for Hepatozoon spp., and 4% for Anaplasma phagocytophilum. PCR testing for Rickettsia spp. and piroplasms was negative. Positive PCR results most likely represent acute infections, which caused no significant clinical manifestations in most cats. IFAT showed 46% of cats had detectable antibodies for Bartonella spp., 30% for Rickettsia spp., and 16% for A. phagocytophilum. Cats were additionally tested for feline coronavirus (antibody ELISA, 60% positive), FIV (antibody-ELISA, 24% positive), and FeLV (antigen ELISA, 0% positive). Many of these VBPs have zoonotic potential and as such have additional importance in human medicine.

Lines 45–47: References to "rabies" and "wildlife" poorly understood. Please, delete it.

Has been deleted.

Lines 50–52: Use pathogen names appropriately.

In our opinion, the names of the pathogens are presented in a reasonable way. May you please specify your comment?

Line 62: The authors state "In general, feline vector-borne pathogens were so far predominantly investigated in domestic cats in Germany". This is in contrast with what was stated in the abstract.

As already mentioned, there is a huge difference in domestic and stray cats according to our opinion.

Lines 64-77: The results of previous studies cannot simply be listed. Sentences of complete meaning must be enunciated in order to include such information and provide the reader with sufficient background to understand where the work that the authors have done is introduced.

In our opinion, it is important to provide the reader with data from domestic cats and wildcats. A comparison of prevalence of vector-borne pathogens between these three different types of “lifestyles” is interesting and therefore, we briefly summarized the data available in literature. We added a short introduction into the topic for clarification.

Line 53 ff: “Stray cats most likely do not have any veterinary care or ectoparasite prophylaxis, and vector contact may be much more frequent than in their domestic counterparts. Previous studies on feline VBPs predominantly focus on domestic cats and may therefore not be representative of the stray population, as VBP prevalence could differ significantly.”

 Lines 79-81: As 64–77

Please see above.

 Line 83: "To the best of the authors’ knowledge", please delete it throughout the manuscript.

Has been deleted.

 Lines 95-96: Not significant for this study.

Has been deleted.

 Lines 104-end of section: The authors cannot limit themselves to listing the tests used by inserting citations. They must accurately explain what they did so that other people can do the same things again.

We added the primers for PCR detection methods (line 97 ff). As commercially available diagnostic assays were used for antibody detection, there was no need for changes in this section in our opinion.

Lines 129-132: Where are these results counted? Why use this test and not a chi square?

Kruskal Wallis is seen as the gold standard if you have more than two groups for comparison. Therefore, we stayed with this test to calculate statistical significance. Shapiro-Wilk is a standard test to look for normal distribution, additionally Kolmogorov-Smirnov testing could be included. However, we did not really see a benefit.

Lines 135-144: Some information is irrelevant.

All the data presented is standard data for describing a study population. We would therefore like to keep all the data presented here.

Table 1: Results regarding the prevalence of FIV, FelV and FIP should not be included in this table, as they represent a risk/predisposing factor and the evidence of exposure to these infections is not the reason for the study.

Has been changed.

Table 2: The table entries "increased" and "decreased" confuse the reader and are not helpful for a better understanding of the table.

We revised the description of the table.

Table 2: “Hematological and biochemical parameters in 50 apparently healthy stray cats in Thuringia (Germany) presenting numbers of cats with elevated and decreased parameters according to the reference interval (RI) by the laboratory in cats tested negative, in cats with coinfections, and in cats tested positive for Hepatozoon spp. and hemotropic Mycoplasma spp. by Polymerase Chain Reaction (PCR)”

 Lines 317-323: MDPI journals do not have a "Limitations of the study" entry. Please explain these limits in the discussion section.

Has been done.

 The discussion is very problematic. The results obtained without a critical view of what was observed and without comparing the results obtained with those of other studies performed in cats, wild felids, or dogs using similar or different approaches. Also, the discussion is very long, and many sentences can be completely eliminated as repetitive. The authors should also discuss the epidemiological role of cats and carnivores in general, which they may have for human infection (it is still not clear to the scientific community whether they are sentinels or reservoirs). Below, some useful works, in my opinion, to improve the discussion.

doi: 10.3390/ani11010009, doi: 10.1016/j.ttbdis.2022.102076, doi: 10.3389/fvets.2021.803424, doi: 10.1016/j.vprsr.2023.100857

Thank you for providing us with some literature. We tried a binary logistic regression analysis, but the model was unfortunately not significant. We carefully revised and reordered the complete discussion according to the reviewer’s suggestions.

Did the authors try to sequence any samples? If not, why hasn't it been done?

This was already stated as a limitation in the manuscript. We were not able to perform sequencing in Hepatozoon spp. as we do not have the possibility to perform this in house. In hemotropic mycoplasma, we were able to differentiate between M. haemofelis, Candidatus M. haemominutum, and Candidatus M. turicensis.

Is other animal data available?

This is also mentioned in the limitations. As the cats included in the study were feral cats, there was not data regarding e. g. living conditions, anamnesis, and ectoparasite prophylaxis.

 Why not carry out a risk factor analysis? What type of sampling or sampling formula was ultimately applied?

We tried to go for binary logistic regression analysis to determine the effect of sex and detection of antibodies for FIV on PCR results. However, the model was unfortunately not statistically significant. We added this to the M&M, results section, and discussion of the manuscript.

Reviewer 2 Report

The article, " Vector-borne infectious pathogens in feral cats in eastern  Germany (Thuringia) including hematology and biochemistry  analysis as well as potential immunosuppression (FeLV, FIV,  feline coronavirus)" describes and thereafter discusses the direct and indirect infection rates of some vector borne diseases, FeLV, FIV and feline coronavirus  from a sample size  of 50 feral cats  presented for sterilization  at a veterinary clinic in the above mentioned region. In addition, biochemistry and hematological parameters were taken from the group and averaged. Although the study is a first report among feral cats in Germany, and adds more epidemiological information in  another resident reservoir population, there are some important limitations to the study and major revision needs to be done before the article should be considered for publication.

Major points:

1.       The sample size of 50 is really very small and at best, would present preliminary data of the  prevalence   of the various diseases discussed in the article, specifically in some areas where as few as 1 cat was captured. How did the authors reach the sample size for statistical significance of the various pathogens in this population?

2.       Although infection rates from the various pathogens differed, some cats which were found not to be infected, they  were analyzed by  hematological and biochemical methods and all results were clumped into a summarized table. What purpose does the table serve unless results were individualized? I would hesitate to add the results unless specific findings in infected cats warrant their inclusion.

3.       The paper is currently not written in sound English style, and it is necessary that a scientifically orientated native English speaker revise and correct the paper.

Minor points:

1.       Materials and Methods: How did the authors prevent capture of the same cats more than once for the sterilization purpose? Did they mark the cats to prevent recapture? If not, were only cats that were previously not sterilized have all the tests performed so as to limit performing the tests on the same cat? Please write this more clearly in the text.

2.       Table 1, line 156: What is the purpose of writing all the following, "; IgM = immunoglobulin M polymerase chain reaction (PCR);  immunofluorescence antibody testing (IFAT); antibody enzyme linked  immunosorbent assay (Ab-ELISA), antigen enzyme linked immunosorbent assay (Ag-ELISA)" in the footnote?

3.       Discussion section:

Some mention of the sensitivities and specificities of the various serological assays should be included in the discussion with mention of their limitations ie crossreactivity.

Line 360: The reference is old. Surely, there is a newer reference.

 The paper is currently not written in sound English style, and it is necessary that a scientifically orientated native English speaker revise and correct the paper.

Author Response

Answer to the editorial and reviewers‘ comments

19-July-2023

Dear Gray, dear reviewers,

thank you for giving us the opportunity to submit a revised draft of our manuscript titled “Vector-borne pathogens in stray cats in eastern Germany (Thuringia)” (ID: animals-2510704) to the Journal “Animals”. We appreciate the time and effort that you and the reviewers have dedicated to providing valuable feedback on our manuscript. We are grateful to the reviewers for their insightful comments, which have improved the paper. We have incorporated changes that reflect all the suggestions provided by the reviewers. All changes are highlighted in red in the revised manuscript. Please see below our point-by-point responses to the reviewers` comments. If any responses may remain unclear or if you wish additional changes, please do not hesitate to let us know.

General comments

According to the suggestions of the academic editor, we changed the term “Feral cat” to “Stray cat” throughout the whole manuscript.

The manuscript was reviewed language wise by a native English speaker, as suggested by reviewer #1 and #2.

We fully agree to both reviewers, that the manuscript needed intense revision and restructuring of the “Result”- and the “Discussion”-sections.

Reviewer #2

The article, " Vector-borne infectious pathogens in feral cats in eastern Germany (Thuringia) including hematology and biochemistry analysis as well as potential immunosuppression (FeLV, FIV, feline coronavirus)" describes and thereafter discusses the direct and indirect infection rates of some vector borne diseases, FeLV, FIV and feline coronavirus from a sample size of 50 feral cats presented for sterilization at a veterinary clinic in the abovementioned region. In addition, biochemistry and hematological parameters were taken from the group and averaged. Although the study is a first report among feral cats in Germany and adds more epidemiological information in another resident reservoir population, there are some important limitations to the study and major revision needs to be done before the article should be considered for publication.

Major points:

  1. The sample size of 50 is really very small and at best, would present preliminary data of the prevalence of the various diseases discussed in the article, specifically in some areas where as few as 1 cat was captured. How did the authors reach the sample size for statistical significance of the various pathogens in this population?

We agree to the fact that the study population included low numbers of stray cats. However, as it is challenging to get samples and data regarding signalment and clinical signs from these cats, we consider our manuscript as valuable, as we present data for the percentage of stray cats tested positive for at least one vector-borne pathogen for the first time in Germany. We also think, that with 50 cats included in the study, a reasonable statistical approach is possible, which is supported by the results of the statistical analysis.

  1. Although infection rates from the various pathogens differed, some cats which were found not to be infected were analyzed by hematological and biochemical methods and all results were clumped into a summarized table. What purpose does the table serve unless results were individualized? I would hesitate to add the results unless specific findings in infected cats warrant their inclusion.

We fully agree to this comment. Therefore, we revised table 2 as well as the “Results” and “Discussion”-section of the manuscript.

  1. The paper is currently not written in sound English style, and it is necessary that a scientifically orientated native English speaker revise and correct the paper.

The language was revised by a native speaker.

Minor points:

  1. Materials and Methods: How did the authors prevent capture of the same cats more than once for the sterilization purpose? Did they mark the cats to prevent recapture? If not, were only cats that were previously not sterilized have all the tests performed so as to limit performing the tests on the same cat? Please write this more clearly in the text.

Has been added.

Line 87 ff: “Only cats with unremarkable general examination findings which had not previously been spayed or neutered were included in the study.”

  1. Table 1, line 156: What is the purpose of writing all the following, "; IgM = immunoglobulin M polymerase chain reaction (PCR); immunofluorescence antibody testing (IFAT); antibody enzyme linked  immunosorbent assay (Ab-ELISA), antigen enzyme linked immunosorbent assay (Ag-ELISA)" in the footnote?

This has been deleted according to the suggestions of reviewer #1.

  1. Discussion section:

Some mention of the sensitivities and specificities of the various serological assays should be included in the discussion with mention of their limitations ie crossreactivity.

The only assay, in which crossreactions may be important, is the A. phagocytophilum IFAT with potential cross reactions with Ehrlichia canis. However, this pathogen is rarely reported in cats and not endemic in Germany yet. We therefore do not include this in the discussion, as it is of no significance for discussion of our results in our opinion.

Line 360: The reference is old. Surely, there is a newer reference.

Has been updated.

Round 2

Reviewer 1 Report

The authors addressed several comments from previous revisions and greatly improved the manuscript, making it clearer, more comprehensive, and more understandable. However, some defects remain, which prevent its acceptance and force me to request major revisions. In particular, the statistical analysis performed is not clear to me (the table has been eliminated, and it is difficult for the reader to follow the results). I advise the authors to include in the manuscript the table relating (contingency table) to the risk factors (sex, age, co-exposure to FIV or FeLV, etc.) including the results of the chosen statistical test. Another weakness is the emphasis placed on the results obtained by testing these cats for FIV and FeLV. The purpose of the manuscript was not to evaluate the prevalence of FIV and FeLV in stray cats in Germany, but this co-exposure is only a possible risk factor. The third critical point is the weight that the haematological and biochemical parameters have in the manuscript (most of the results, a table, some graphs to indicate results that are almost never statistically significant). I am sure that the authors can solve these problems, as the work is potentially publishable for the data it presents (it is necessary to improve its presentation). Here are some minor comments.

Line 19: "Are predisposed to" could be better.

Line 22 (Also 86): It is not clear what "TNR program" is for readers. Please delete it (and also "Cats with unremarkable general examination were included"). It is not necessary to describe under what plan or how these animals were sampled in the abstract (this information is then repeated in the Materials and Methods section).

Lin 29-30: See "Another weakness is the emphasis placed on the results obtained by testing these cats for FIV and FeLV. The purpose of the manuscript was not to evaluate the prevalence of FIV and FeLV in stray cats in Germany, but this co-exposure is only a possible risk factor." Readers are only interested in whether co-exposure to FIV and FeLV are predisposing factors to other infections, not how much FIV and FeLV is circulating in Germany.

Lines 30-31: This sentence must be improved.

Line 69: "One" in "one"

Lines 74-77: It has no significance at this point in the manuscript. Delete, could be used in the discussion to comment on the result.

Line 136-137: Where is the binary logistic regression? Please, describe what was performed for the statistical analysis.

Figure 1: As I said in the general comments, I don't understand why you devote a large part of the manuscript to describing these insignificant results. The authors should think about representing these results in an appropriate graphic format and perhaps inserting them as supplementary material.

Line 240: FIV western blot?

Lines 247-249: Please delete it.

Line 261-263: How were the animals clinically evaluated? Have special visits been made? Being stray animals, there is no history of these animals. How do the authors know that they were healthy animals? It would be more appropriate to say that the positive animals showed, however, the disease in an asymptomatic form (which is also common in other species, see foxes).

Line 267: The authors discuss a study in which high prevalences were obtained in feral cats using the spleen as a sample. So can the matrix used affect these prevalences?

Line 362: The limitations of such a study should also be described here. Although the authors investigated numerous pathogens, the sample of 50 animals is certainly not representative and should be expanded for more efficient surveillance of these infections.

Lines 365-368: Please, delete it.

Discussion: Despite my advice, the authors argue their data against a couple of studies. My advice is to summarize the discussion and add the comparison between feral cats/feral dogs and feral cats/wild animals (foxes, wolves, etc. in addition to the feral cats that the authors have already entered). Greater emphasis should be given to the role that carnivores play in being reservoirs and/or sentinels of these infections, given that, at least some species, would appear to be almost asymptomatic (eg Anaplasma in several animal species). What is the point of having identified molecular and serological data if they are not compared with those obtained in other susceptible species from other countries? I renew those publications that can be useful for this comparison. doi: 10.3390/ani11010009; doi: 10.1016/j.ttbdis.2022.102076; doi: 10.3389/fvets.2021.803424; doi: 10.1016/j.vprsr.2023.100857; https://doi.org/10.1186/s13028-023-00699-6

English has improved significantly since the revision.

Reviewer 2 Report

The paper, which presents new information on VBDs in cats in a region of Germany, has been moderately improved but certain shortcomings are still evident and need to be addressed before it is considered for publication.

The major points are the small  sample size which would limit this publication rather to a short communication in a journal of regional importance. English needs still to be improved and reorganization of the M&M, Results and Discussion sections is mandatory as they are very difficult to follow and not organized in a targeted structure currently.

Some examples follow:

Line 36: "importation of dogs and cats from abroad"

Line 54: "exposure " instead of "contact"

Line 80: "stray cats caught for the purpose of  castration/sterilization"

Line 88: "unremarkable findings on general examination and which had not previously  been …"

Line 85: The M&M section needs to be rewritten as it is not sufficient in its current state. It is not necessary to list each set of primers for each reaction if there is a reference for the PCR. Rather state, "as performed by….(citation)". Alternately, a Table with the list of primers and reaction conditions can be added and the author of the original publications added in another column.

Line 145: Please rewrite, "Wildcats were excluded from this study due to the  signalment". What does this mean? Was it their phenotypic properties that excluded them from the study?

The "Results" section should be rewritten. Try list the main findings first and then in a focused way, moving towards other findings observed. Rather refer to a Table of results then list long paragraphs of the specific findings. This confuses the reader and is less illustrative. I still do not understand the purpose of listing the average hematological findings unless comparing specific cats with vector borne disease agents and the specific biochemical or hematological finding.   

Again, in the Discussion section, rewrite listing the main findings of the paper and then targeted towards "take home message/es". It does not strengthen the paper when the first chapter already states some of the study's limitations and shortcomings. This should be listed last.

English still needs to be improved

Round 3

Reviewer 1 Report

The authors have addressed my comments in this round of reviews. The manuscript is very close to acceptance; below are some of my minor comments.

Abstract:

Line 29: "This did not have an impact on test results." could change to "We didn’t find any correlation between FIV-FelV-FIP status and VBP positivity" or something similar.

Line 31: Authors should put the word "surveillance" in the sentence. The surveillance of these infections, both in humans and animals, is the real task of the One Health concept.

Discussion:

Line 307–312: This sentence could be deleted as it is repetitive and adds no information to the discussion (also, the authors again report FIV western blot).

Only minor corrections are needed.

Reviewer 2 Report

The paper has been significantly improved. I would still have suggested its publication as a short report because of the small sample group.

Minor changes:

Line 96: Please explain clearly what indirect detection methods  are as it is not clearly stated that IFAT was an indirect method here.

Table 1 is still not complete. Add another column with a reference number referring to a publication that used the same primers and  the conditions used in each PCR reaction.  

Line 1645: Don't italicize regression

 Line 100: Remove "each"

Worthy of a revision again but significantly improved
